# fmMAP: A Framework Reducing Site-Bias Batch Effect from Foundation Models in Pathology

**Hai Cao Truong Nguyen**                                          HAINCT@NCC.RE.KR
*Department of Public Health and AI*
*Graduate School of Cancer Science and Policy*
*National Cancer Center, Goyang, Republic of Korea*

**David Joon Ho**                                                 HOD@NCC.RE.KR
*Department of Public Health and AI*
*Graduate School of Cancer Science and Policy*
*National Cancer Center, Goyang, Republic of Korea*

## Abstract

Foundation models (FMs) in pathology are general-purpose models capturing heterogeneous morphological patterns on pathology images leveraged by a vast training dataset. Although FMs have demonstrated promising results in multiple downstream tasks such as classification and retrieval, confounding factors are also embedded in the features potentially causing inaccurate decisions. For example, we observe a batch effect where distinctive medical center signatures are displayed when clustering features from FMs. In this work, we propose Foundation Model-based Manifold Approximation Pipeline (fmMAP) to reduce the batch effect by adjusting features from FMs. Our framework employs supervised uniform manifold approximation (UMAP) to transform features generated by FMs into an optimal space. In this transformed space, characteristics of features of interest (i.e., biological features) are highlighted while other confounding factors are reduced. Experimental results on eight recent FMs show that raw features from the FMs are shown to be unrobust, but fmMAP transforms features to become robust on all FMs according to the robustness index. In addition, fmMAP reduces average balanced accuracy for site prediction and improves average balanced accuracy for tissue type classification achieving more than 96% in publicly available datasets. We expect fmMAP framework will help FMs identify essential pathologic features that would enhance performance on downstream tasks. The code is available at `https://github.com/davidholab/fmMAP`.

**Keywords:** foundation model, batch effect, image classification, histopathology

## 1 Introduction

Foundation models (FMs) are general-purpose machine learning models conducting a wide range of downstream tasks (Bommasani et al. (2021)). FMs in pathology trained on millions of whole slide images (WSIs) in a self-supervised manner can capture heterogeneous morphological patterns on pathology images (Waqas et al. (2023)). For example, several FMs in pathology have demonstrated promising results in patch-level classification, slide-level classification, patch retrieval, and biomarker prediction (Chen et al. (2024); Filiot et al. (2023, 2024); Saillard et al. (2024); Vorontsov et al. (2024); Xu et al. (2024); Zimmermann et al. (2024)).

While representations from FMs can be used in several downstream tasks, they may be suffered from confounding factors. For example, we observe a site-bias batch effect

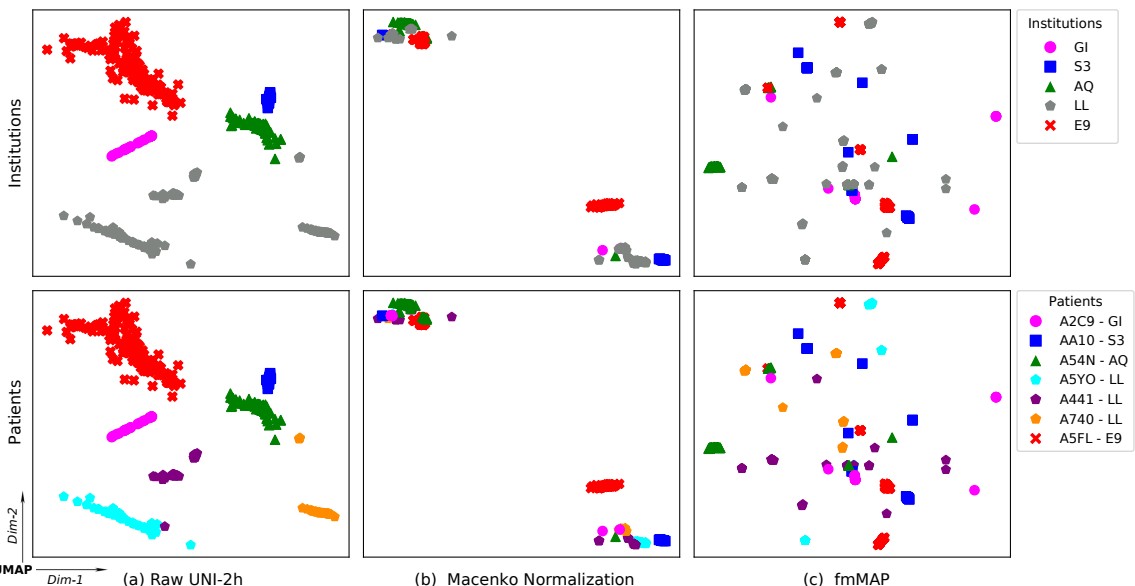

Figure 1: A UMAP (McInnes et al. (2018)) illustration of the site-bias effect in a subset of Breast Cancer Semantic Segmentation (BCSS) Dataset (Amgad et al. (2019)). (a) Raw features extracted from FM UNI-2h (Chen et al. (2024)) clearly show a site-bias batch effect. Furthermore, the patient-bias batch effect can also be observed (institution: LL, patients: A5YO, A441, A740). (b) Macenko stain normalization (Macenko et al. (2009)) is applied to patches before feature extraction. The results show that stain normalization does not effectively reduce the batch effect in both cases. (c) After applying our proposed fmMAP, the batch effect is significantly reduced in both cases. Samples are distributed relatively evenly throughout the space and tend to cluster by tissue type rather than by the source site. More analysis results from other FMs are available in Appendix B for reference.

in these FMs. In Fig. 1(a), patches are clustered sharply by institutions. This batch effect is known to be caused by different staining protocols and scanners among different institutions (Stacke et al. (2021)). A common technique to overcome this batch effect is stain normalization (Macenko et al. (2009); Reinhard et al. (2001)), but the same limitation remains, as shown in Fig. 1(b). The same issue was recently presented (de Jong et al. (2025); Kömen et al. (2024)), confirming that confounding factors are embedded within the features extracted by FM. The site-bias batch effect could lead to biased predictions, potentially raising ethical concerns (Howard et al. (2021)). While the site-bias batch effect caused by FMs was discussed (de Jong et al. (2025); Kömen et al. (2024)), no solution has yet been suggested.

In this paper, we propose fmMAP, Foundation Model-based Manifold Approximation Pipeline, a framework to significantly reduce the site-bias batch effect by adjusting features from FMs, shown in Fig. 1(c). To the best of our knowledge, fmMAP is the first approach to remove batch effect from FMs in pathology. The fmMAP framework includes modifications to mitigate common confounding biases in patch-level images, such as variations in staining, before extracting features using FMs. Then, fmMAP uses UMAP (McInnes et al. (2018)) to

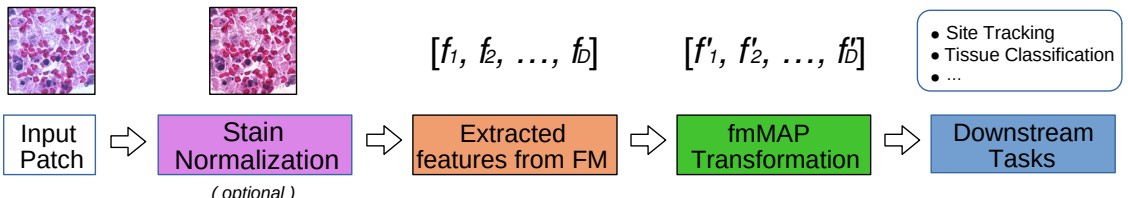

Figure 2: Workflow of fmMAP framework.

transform the extracted features from FMs into an optimal space, where biological features are greatly enhanced while confounding features are suppressed. We test fmMAP on eight FMs and two public datasets, where fmMAP diminishes site tracking and improves tissue classification on all FMs. Specifically, all post-fmMAP features demonstrate that they are greater than the "robust" threshold (Kömen et al. (2024)). In addition, the average balanced accuracy of eight FMs even exceeds 96% on tissue classification. Our results suggest that the fmMAP framework will help FMs recognize crucial pathological characteristics and improve their performance in downstream tasks.

## 2 Proposed Method

Recent FMs have shown great promise in pathology. Still, their performance is often reduced by confounding batch effects introduced by variations in staining protocols, imaging conditions, and site-specific differences (Stacke et al. (2021)). Although stain normalization methods, such as Macenko et al. (2009) and Reinhard et al. (2001), have been widely used to address this challenge, we observe that these methods do not improve FM's performance, as shown in de Jong et al. (2025); Kömen et al. (2024).

Standard domain adaptation techniques such as Correlation Alignment for Unsupervised Domain Adaptation (CORAL) (Sun et al. (2016)) or Adversarial Discriminative Domain Adaptation (ADDA) (Tzeng et al. (2017)) may be inadequate for the ultimate goal of improving the performance of downstream analysis. CORAL is unsuitable for FMs due to its reliance on second-order statistics (covariances), linear transformations, and computational inefficiency in high dimensions. Similarly, ADDA is limited by training instability, loss of task-relevant information, high computational cost, and poor generalization across multiple batches. Both methods are designed for simpler domain adaptation scenarios and struggle with the scale, complexity, and heterogeneity of FMs. It is necessary to have a scalable workflow applying a non-linear estimation implemented directly on the features of any FMs.

This highlights a more robust approach that preserves meaningful biological information while effectively reducing batch effects. We propose fmMAP to integrate stain normalization with a UMAP-based transformation for optimizing feature representations, offering pathology FMs reliable and generalizable performance across diverse datasets. Figure 2 depicts the fmMAP framework.

The proposed fmMAP leverages a modified version of UMAP, known as "supervised UMAP", which incorporates target categories (labels) (McInnes et al. (2018)). However, unlike neural network training, supervised UMAP does not fuse labels directly into features, which may distort the feature content. Labels are used to build a fuzzy graph of connectiv-

ities between samples to assign a higher priority to samples within a category. Instead of finding a projected graph in a lower-dimensional space, fmMAP searches for a projection of that graph in the *"same-dimensional space"*. As a result, this will not shorten the size of the input feature or reduce the capacity of the content, preserving as much original information as possible. Therefore, fmMAP will not suffer from low-dimensional distance measurements of UMAP, but it preserves and enhances the robustness of FM features. We use the "robustness index" from de Jong et al. (2025) to support this point. In addition, this approach would serve as a denoising procedure to mitigate confounding factors, given the emphasis on labels, while maintaining both local and global properties of features through the graph structure.

## 2.1 Unsupervised UMAP

Assume that UMAP models a high-dimensional dataset $X$ with $N$ samples as a weighted graph. We have $X = \{x_1, x_2, \ldots, x_N\} \subset \mathbb{R}^D$, where each $x_i$ is a data point described by a $D$-dimensional attribute vector. The pairwise distances $d_{ij}^X$ between two features (data points) $x_i, x_j$ can be converted to conditional probabilities:

$$p_{ij} = \exp\left(-\frac{d_{ij}^X - \rho_i}{\sigma_i}\right) \tag{1}$$

where $\rho_i$ is the distance to the nearest neighbor to enforce local connectivity, and $\sigma_i$ is a scaling parameter determined by the k-nearest-neighbor distances. This forms the unsupervised component representing the local manifold structure, which is the fuzzy graph $G^D$, in high-dimensional space. In low-dimensional space, UMAP searches for embeddings $Y = \{y_1, y_2, \ldots, y_n\} \subset \mathbb{R}^d$, $d \ll D$ for dimension reduction. Assume $q_{ij}$ is the similarity distribution between two embedded points $y_i, y_j$, conditional probabilities $q_{ij}$ form the fuzzy graph $G^d$ in low-dimensional space. The unsupervised objective function optimizes the cross-entropy loss $L_{\text{unsup}}$ between $p_{ij}$ and $q_{ij}$:

$$L_{\text{unsup}} = -\sum_{i \neq j} p_{ij} \log q_{ij} \tag{2}$$

## 2.2 Supervised UMAP

To incorporate the target categories, assume $C = \{c_1, c_2, \ldots, c_N\}$ is the corresponding class labels of dataset $X$. A simple label similarity function can be defined as:

$$d_{ij}^C = \begin{cases} 0 & \text{if } c_i = c_j \\ 1 & \text{if } c_i \neq c_j \end{cases} \tag{3}$$

Instead of $d_{ij}^X$, the supervised pairwise distance function is modified to account for both feature and label similarity:

$$d_{ij}^{\text{sup}} = \alpha \cdot d_{ij}^X + (1 - \alpha) \cdot d_{ij}^C \tag{4}$$

where $\alpha \in [0, 1]$ controls the influence of label supervision. Equation (1) will become:

$$p_{ij}^{\text{sup}} = \exp\left(-\frac{d_{ij}^{\text{sup}} - \rho_i}{\sigma_i}\right) \tag{5}$$

And, the supervised objective loss function will be:

$$L_{\text{sup}} = -\sum_{i \neq j} p_{ij}^{\text{sup}} \log q_{ij} \tag{6}$$

From equations (3-6), it can be observed that $d_{ij}^C$ are constants and class labels only influence the graph construction, not the optimization stage. This is the best part of supervised UMAP compared to other feature-fusing or feature-concatenating approaches. The original feature content is preserved during optimization. That helps preserve category structures in the final embeddings, while the blended distances $d_{ij}^{\text{sup}}$ encourage intra-class clustering and inter-class separation.

## 2.3 Experimental Setting

Eight recent FMs are selected to join our experiments, including H_Optimus_0 (Saillard et al. (2024)), Phikon (Filiot et al. (2023)), Phikon-v2 (Filiot et al. (2024)), Prov-GigaPath (Xu et al. (2024)), UNI (Chen et al. (2024)), UNI-2h (Chen et al. (2024)), Virchow (Vorontsov et al. (2024)), and Virchow2 (Zimmermann et al. (2024)). These FMs and their upgraded versions are selected for high performance in a previous study (Kömen et al. (2024)). Their diverse training data ensures varied data coverage testing. As Virchow and Virchow2 combine class and mean patch tokens for the final tile embedding, we also test the variants with 1280 class tokens for a fair comparison with other methods. The '_1280' tag denotes these Virchow-related FM variants.

For additional details on the implementation of the proposed methods, such as hyperparameters, datasets, stain normalization, and downstream tasks, see Appendix A.1, A.2, A.3, A.4, respectively.

## 2.4 Evaluation Metric

Robustness index of an FM is proposed to measure whether a set of biological features dominates a set of confounding features (de Jong et al. (2025)). The robustness index is defined as the number of neighbors with the same biological class (e.g., tissue type) and the number of neighbors with the same medical center within the $K$ nearest ones in an embedding space:

$$R_K = \frac{\sum_{i=1}^{N} \sum_{j=1}^{K} \delta_{b_i, b_j}}{\sum_{i=1}^{N} \sum_{j=1}^{K} \delta_{m_i, m_j}} \tag{7}$$

where $b_i$ and $b_j$ denote the label of the biological class of the $i$-th sample and its $j$-th nearest neighbor, respectively; $m_i$ and $m_j$ represent the medical centers of the respective samples; Kronecker delta function $\delta_{xy}$ returns 1 if $x$ and $y$ are equal, and 0 otherwise; and $N$ is the number of samples. The numerator counts the nearest neighbors sharing the same biological class, while the denominator counts those from the same medical center.

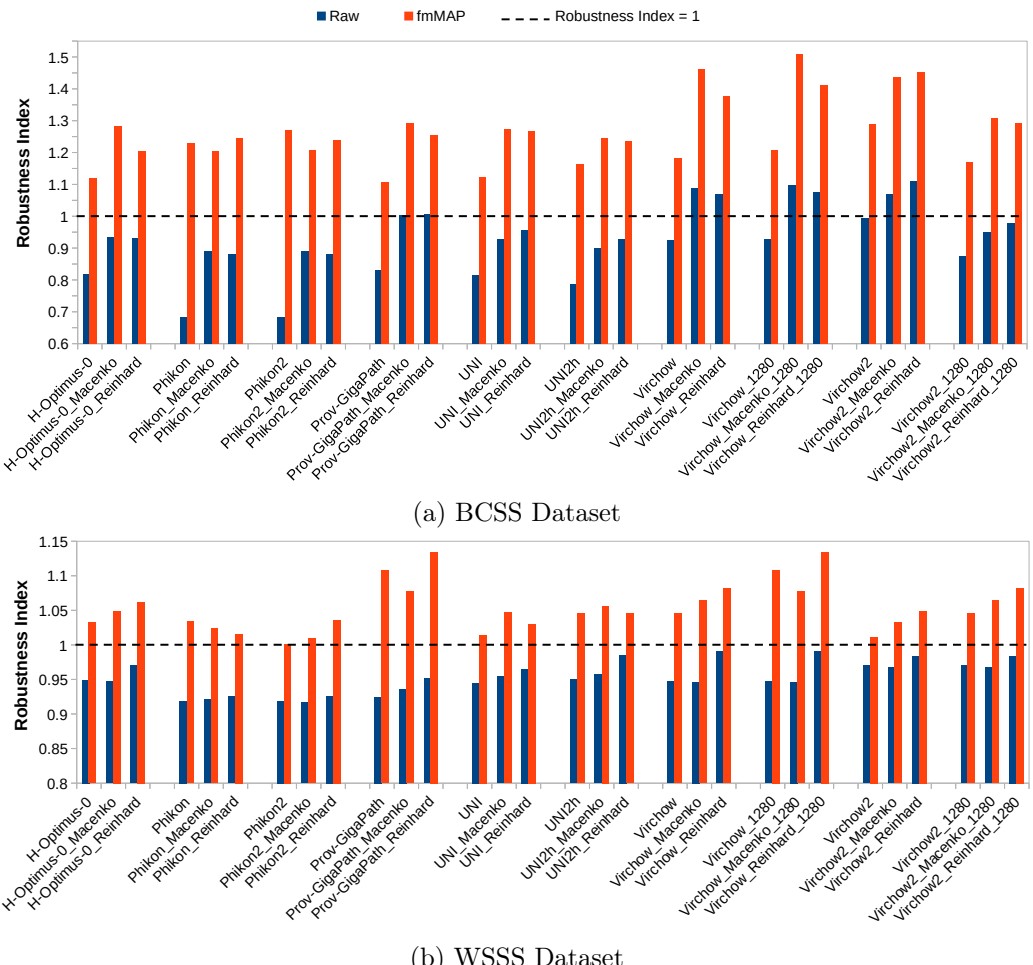

(a) BCSS Dataset

(b) WSSS Dataset

Figure 3: Robustness indices of foundation models (FMs) and their variants on two datasets before and after fmMAP. Raw features from FMs achieve robustness indices less than 1. fmMAP transforms the features to achieve robustness indices greater than 1.

The cosine distance is used as the similarity metric between embeddings. $R_K \gg 1$ would indicate that an embedding space is structured primarily by biological signals rather than confounders, whereas $R_K \ll 1$ would indicate the opposite. We use the robustness index $R_K = 1$ as a threshold to identify "robust" FMs. In this study, we set $K = 50$ used in de Jong et al. (2025).

Please see Appendix A.5 for the definition and calculation of the average balanced accuracy (ABA).

## 3 Experimental Results

### 3.1 fmMAP improves the robustness of FMs

Figure 3 shows that fmMAP significantly improves the robustness of all FMs in both datasets. Without fmMAP, none of the FMs using raw features are classified as robust,

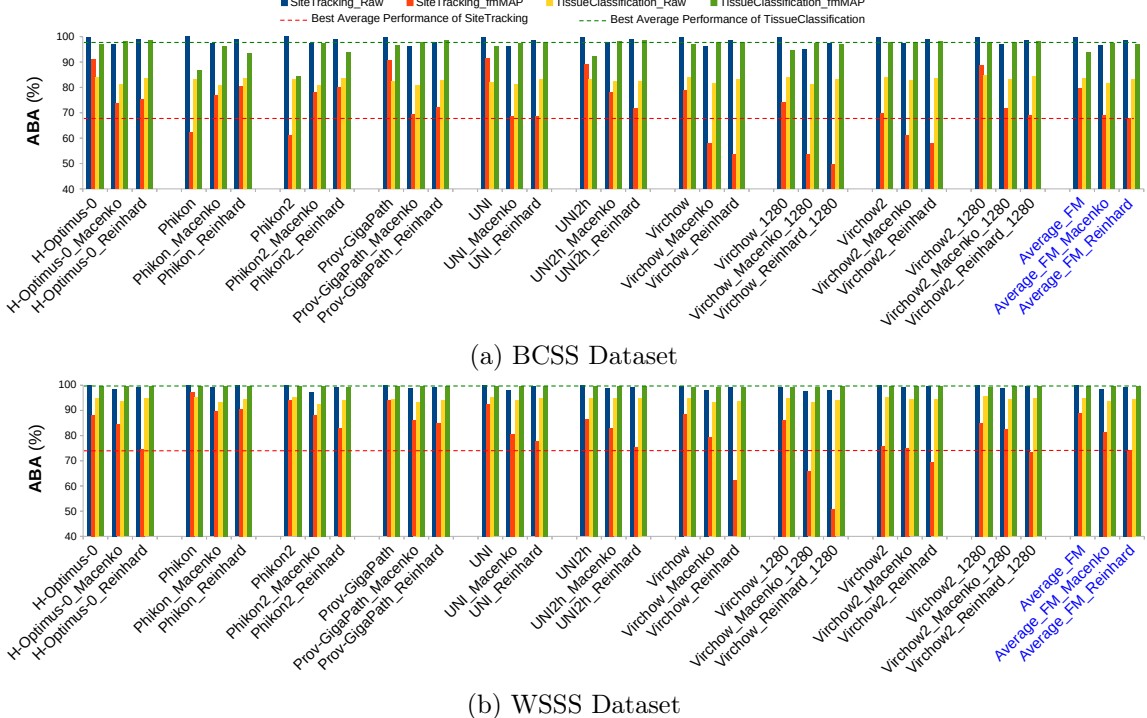

(a) BCSS Dataset

(b) WSSS Dataset

Figure 4: Average balanced accuracy (ABA, details in Appendix A.5)) performance of foundation models (FMs) and their variants on two datasets. Each FM comes with two pairs of bars indicating ABA of site tracking and tissue classification, with and without fmMAP for each. ABA for site tracking is consistently and significantly reduced, and ABA for tissue classification shows a notable and appropriate increase. This pattern is evident across all FMs on both datasets, demonstrating fmMAP's critical role in boosting the overall performance of FMs. The average performance over all FMs is added to highlight the general trend. Dash lines indicate models with the best performances among the average ones.

$R_K < 1$, generally consistent with results reported in (de Jong et al. (2025)). Across both datasets, stain normalization slightly enhances the robustness of all FMs. However, in most cases, this improvement is insufficient to exceed the robustness threshold, $R_K = 1$. All FMs and their variants exhibit increased robustness after applying fmMAP, as indicated by the robustness index, $R_K > 1$. For example, Virchow with Macenko normalization and 1280 class tokens achieves the highest robustness index, $R_K = 1.51$. We notice that the extent of robustness improvement varies between Breast Cancer Semantic Segmentation (BCSS) Dataset (Amgad et al. (2019)) and Weakly Supervised Semantic Segmentation for Lung Adenocarcinoma (WSSS4LUAD) Dataset (Han et al. (2022)), which may be caused by the different numbers of medical centers in the two datasets.

## 3.2 fmMAP improves the accuracy of FMs

Figure 4 displays ABA derived from a stratified 5-fold cross-validation test. For each FM, we report ABA results for predicting source sites and classifying tissue types, both with

and without fmMAP applied. Implementing fmMAP remarkably lowers ABA for source site prediction. The most substantial drops, around 47%, occur in BCSS dataset in two Virchow variants: Virchow_Reinhard_1280 and Virchow2_Reinhard_1280. Conversely, tissue classification ABA shows significant gains. Especially, in BCSS dataset, no FM initially exceeds ABA of 85% before fmMAP. After fmMAP, however, all FMs surpass 96%, except Phikon, Phikon-v2, and their variants. This consistent improvement holds across both BCSS dataset and WSSS4LUAD dataset.

### 3.3 fmMAP compensates for the over-adjustment of stain normalization

Figure 4 additionally shows that while stain normalization such as Macenko (Macenko et al. (2009)) or Reinhard (Reinhard et al. (2001)) alone drops ABA for site tracking, it also drops ABA for tissue classification. For example, in the case of Hoptimus0 in Figure 4(a), ABA for site tracking decreases from 99.6% to 96.8% after Macenko, but ABA for tissue classification also decreases from 83.9% to 81.3% after Macenko. This pattern is consistent with previous studies (de Jong et al. (2025); Kömen et al. (2024)). Figure 4 further reveals that applying fmMAP with stain normalization consistently elevates ABA to the highest levels seen among the variants. This suggests that fmMAP and stain normalization can cooperate well to improve the general performance of FMs.

## 4 Conclusion and Future Work

In this study, we find that pathology FMs are significantly affected by confounding factors such as site-specific signatures. Stain normalization methods can degrade the biological signal within extracted features and are insufficient to eliminate batch effects. We introduce fmMAP, a framework designed to enhance FM robustness. Experiments show that our approach can effectively remove site-specific biases while strengthening biological signals from FMs.

One limitation of our work is that this would require patch-level labels. When labels are unavailable, one potential solution that we would suggest is to use the raw features generated by FMs to predict slide-level labels. Since FMs generally excel at generating input representations without requiring patch labels, we can take exhaustive advantage of this to predict the slide label first. Later, these labels can be used for fmMAP to enhance downstream information. This proposes an efficient way for the fmMAP framework to undermine the generality of the FM representations. We will also test the ability of fmMAP to handle multiple biases simultaneously, mentioned in the Appendix A.6. Another plan to further investigate fmMAP is to determine the minimum number of patch-label pairs that are sufficient for downstream analyses, thus reducing the annotation burden on pathologists. Lastly, we plan to test fmMAP in other downstream tasks such as slide-level classification, image retrieval, and biomarker prediction.

### Acknowledgments and Disclosure of Funding

This work was supported by the National Cancer Center Grant (NCC-24H1730-2). The authors declare no competing interests.

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

## Appendix A. Supplementary Materials

### A.1 Hyperparameters of UMAP

Uniform Manifold Approximation and Projection (UMAP) (McInnes et al. (2018)) is a technique in machine learning that reduces complex, high-dimensional data into a lower-dimensional form. A graph is built in the high-dimensional space of features, and another graph will be estimated in a low-dimensional space. The method strikes a balance between preserving nearby relationships (local structure) and the global data structure. fmMAP adopts this mechanism of UMAP to search for the optimal projection of FM features in the manifold, but under the additional guidance of the labels.

The UMAP algorithm takes four hyperparameters: $n\_neighbors$, the number of neighbors to consider when approximating the local metric; $n\_components$, the target embedding dimension; $min\_dist$, the desired separation between close points in the embedding space; and $n\_epochs$, the number of training epochs to use when optimizing the low-dimensional representation (McInnes et al. (2018)). Hyperparameters allow UMAP to work adaptively with a wide range of data with various characteristics. However, we agree that tuning the model costs a significant amount of time in a bad case.

As mentioned in Section 2, since the purpose of fmMAP is not dimension reduction, fmMAP will always use the $n\_components$ the same as the size of the input features, i.e., $d = D$ for $X$ and $Y$ described in Section 2.1. The $n\_epochs$ is automatically optimized by the Python package 'umap' according to the data size. We use the default values of that package for $n\_neighbors$=15, the target embedding dimension; $min\_dist$=0.1 for the other two hyperparameters. The default setting works well for the selected test datasets. In Equation 4, we use $\alpha$=0.5 to balance the contribution between features and labels. However, users can still adjust these parameters to match their settings.

### A.2 Datasets

In this study, we use two multi-institutional public datasets with pixel-level or patch-level labels: Breast Cancer Semantic Segmentation (BCSS) Dataset (Amgad et al. (2019)) and Weakly Supervised Semantic Segmentation for Lung Adenocarcinoma (WSSS4LUAD) Dataset (Han et al. (2022)). First, BCSS dataset contains 151 pathology images with pixel-level labels, which are sourced from The Cancer Genome Atlas (TCGA). The dataset has five classes: tumor, stroma, lymphocytic infiltration, necrosis, and other tissue types. We assign one of the classes to patches if the class is annotated at least 70% of its area, ensuring that the assigned labels closely represent their predominant content. At the end, we have 19,337 patches from 20 medical centers, providing a broad spectrum of histopathological variations and confounding factors, making it well-matched to the objectives of our experiments.

Second, a training set of WSSS4LUAD dataset is collected from two sources, containing 49 images from Guangdong Provincial People's Hospital (GDPH) and 14 images from TCGA, providing 10,091 patches with patch-level labels. The dataset provides three tissue type labels: tumor, stroma, and normal. We assign final labels to patches by prioritizing labels by medical significance, with tumor over stroma and stroma over normal tissue. Note that we did not include validation and test sets in this study because weak labels are not provided.

All patches are extracted at 20× magnification with a size of 224×224 pixels. Since our datasets are highly imbalanced, we find that stratified five-fold cross-validation is appropriate to assess the performance of the fmMAP framework. The final performance is determined by averaging the results obtained from five iterations.

## A.3 Stain Normalization

Stain normalization is essential in digital pathology, as it minimizes differences caused by varying staining techniques and imaging setups. Macenko approach (Macenko et al. (2009)) uses the singular value decomposition to determine the covariances of the stain color and standardize histological images. Reinhard method (Reinhard et al. (2001)) adjusts the color profile of one image to align with another by matching statistical properties in a color space that reflects human perception. Both techniques successfully decrease variability between samples. fmMAP adopts stain normalization as a required preprocessing step for input patches before feeding the FMs. Any future advances in this field can be adaptively integrated or exchanged to improve the whole performance.

## A.4 Downstream Tasks

We adopt a linear probe test tool provided by UNI (Chen et al. (2024)) to perform two downstream tasks: site source tracking and tissue classification. Source site tracking aims to assess the presence of site-specific signatures in FMs. If a source site can be tracked, this would suggest that the features retain distinct site-related information strong enough to reveal their origins. We conduct this test before and after applying fmMAP. Our two goals are to demonstrate that fmMAP effectively mitigates batch effects in FMs and to evaluate the influence of confounding factors on tissue classification captured in the extracted features. Since morphology is the primary information expected from FMs, we investigate how site-specific content impacts the biological signal within FM features. Analyzing metric changes before and after applying fmMAP, we assess its effectiveness in achieving improved downstream tasks. Positive results would indicate that removing batch effects helps restore morphological information that was previously masked.

## A.5 Average Balanced Accuracy

Due to a significant imbalance in the number of samples between medical sites and uneven distributions of tissue classes, the standard accuracy metric may be biased toward the dominant category. We use average balanced accuracy (ABA) (Sokolova and Lapalme (2009)) as our metric to evaluate performance results instead of standard accuracy, defined as:

$$ABA = \frac{1}{C} \sum_{c=1}^{C} Acc_c \tag{8}$$

where $Acc_c$ is the accuracy for class $c$ and $C$ is the number of classes.

### A.6 fmMAP can mitigate multiple types of bias simultaneously

The underlying assumption of fmMAP is that a fixed size of representations has a limited capacity to encode all the information within an image, including biological morphology, source site, patient, staining, and possibly other factors. Since the information capacity remains fixed, by adjusting to maximize the proportion of the desired information of interest (biological morphology), we would expect the relative contributions of the other types of information, considered as biases, will naturally be diminished. Consequently, this approach may potentially reduce other biases. Furthermore, such a mechanism in that paradigm should willingly accommodate the overfitting.

For example, Fig. 1 illustrates the UMAP projections of UNI-2h (Chen et al. (2024)) features of a subset from BCSS dataset, before and after applying fmMAP. Fig. 1(a) clearly shows the site-bias batch effect among institutions (top), as well as the inter-person batch effect among patients within the same institution (bottom). It can be observed in Fig. 1(b) that stain normalization does not properly reduce the batch effect in both cases. However, in Fig. 1(c), the batch effect is significantly reduced after applying fmMAP in both cases. The samples are distributed fairly equally throughout the space and tend to gather according to tissue type instead of source site or patient. This pattern can also be consistently observed in the visualized results of other FM tests in the Appendix B.

### Appendix B. Supplementary Figures

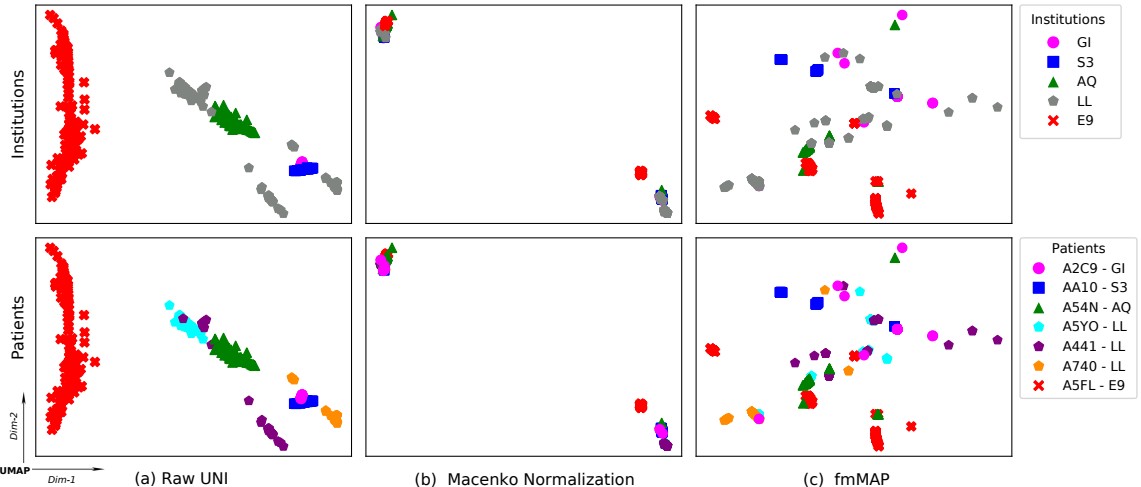

Figure B.1: A UMAP illustration of site-bias effect in a subset of BCSS dataset using FM UNI (Chen et al. (2024)). (a) Raw features extracted from FM clearly show the site-bias batch effect and patient-bias batch effect. (b) Macenko stain normalization is applied to patches before feature extraction, but the effect is very limited. (c) After applying fmMAP, the batch effect is significantly reduced in both cases.

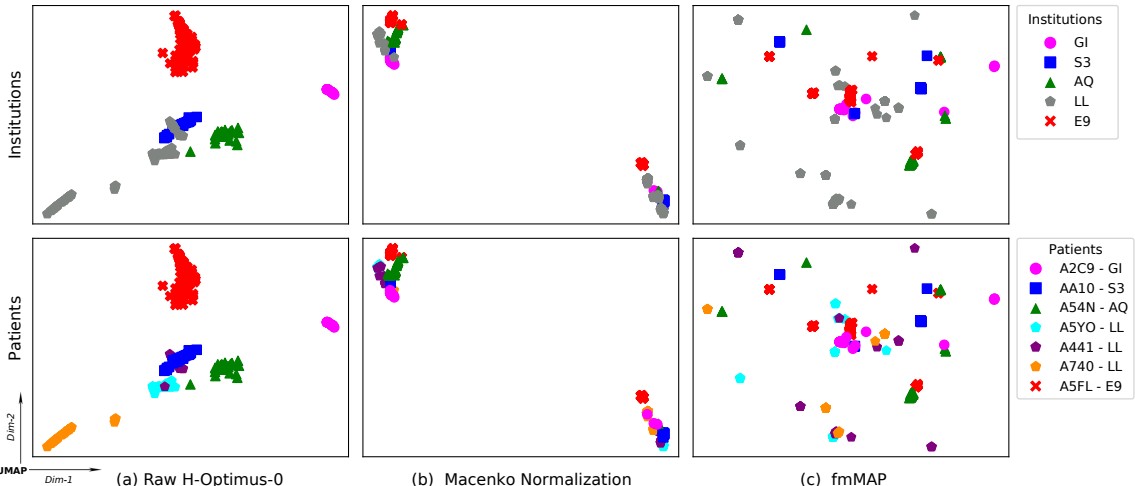

Figure B.2: A UMAP illustration of site-bias effect in a subset of BCSS dataset using FM H-Optimus-0 (Saillard et al. (2024)). (a) Raw features extracted from FM clearly show the site-bias batch effect and patient-bias batch effect. (b) Macenko stain normalization is applied to patches before feature extraction, but the effect is very limited. (c) After applying fmMAP, the batch effect is significantly reduced in both cases.

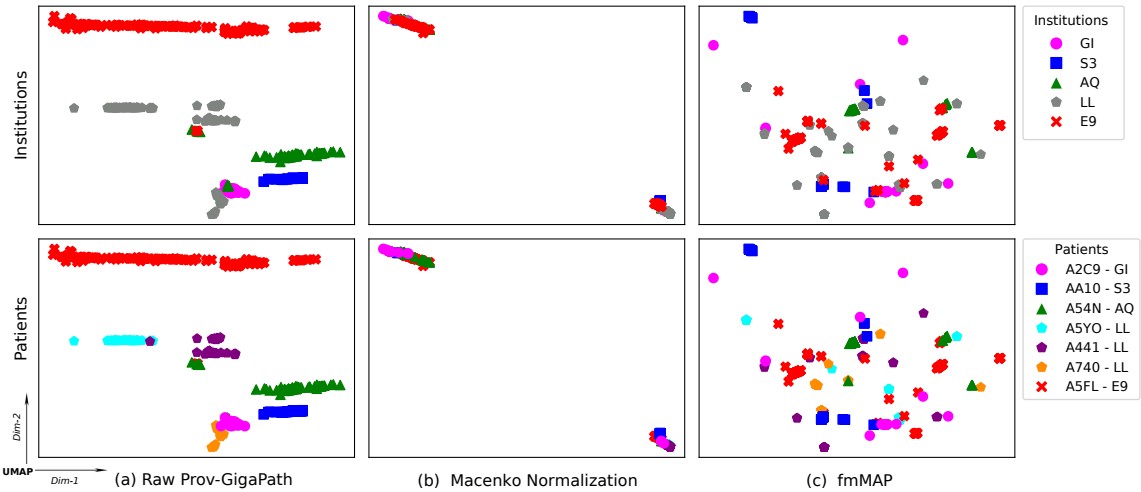

Figure B.3: A UMAP illustration of site-bias effect in a subset of BCSS dataset using FM Prov-GigaPath (Xu et al. (2024)). (a) Raw features extracted from FM clearly show the site-bias batch effect and patient-bias batch effect. (b) Macenko stain normalization is applied to patches before feature extraction, but the effect is very limited. (c) After applying fmMAP, the batch effect is significantly reduced in both cases.

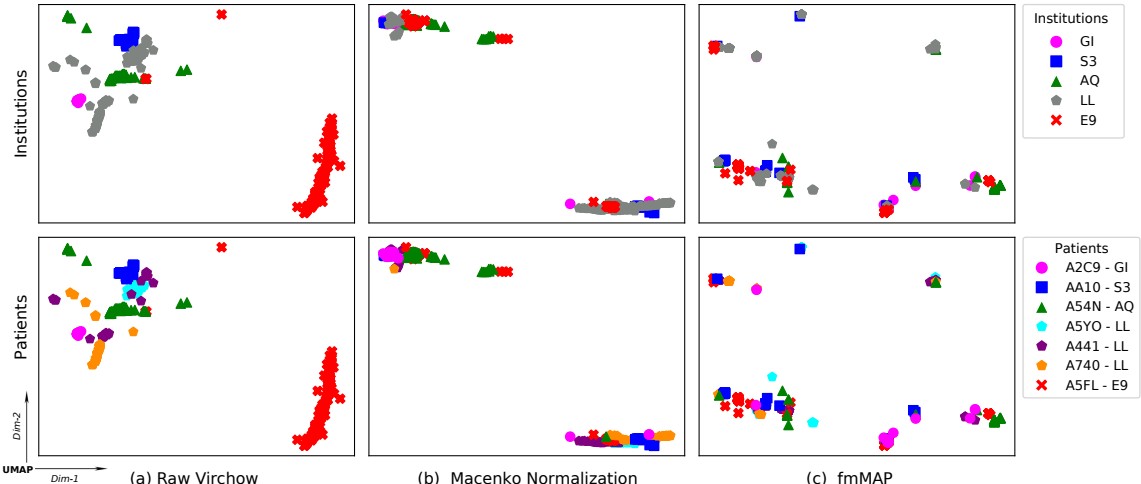

Figure B.4: A UMAP illustration of site-bias effect in a subset of BCSS dataset using FM Virchow (Vorontsov et al. (2024)). (a) Raw features extracted from FM clearly show the site-bias batch effect and patient-bias batch effect. (b) Macenko stain normalization is applied to patches before feature extraction, but the effect is very limited. (c) After applying fmMAP, the batch effect is significantly reduced in both cases.

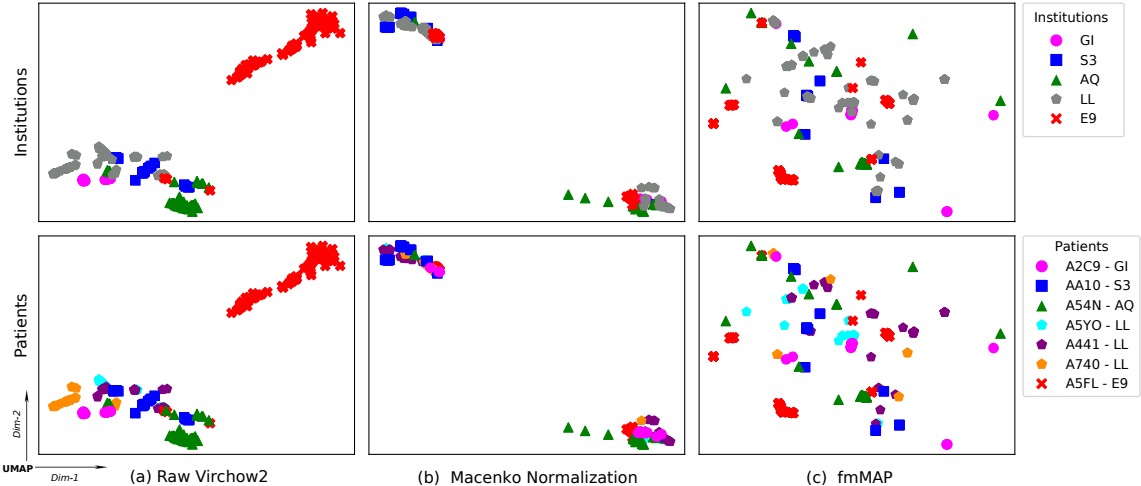

Figure B.5: A UMAP illustration of site-bias effect in a subset of BCSS dataset using FM Virchow2 (Zimmermann et al. (2024)). (a) Raw features extracted from FM clearly show the site-bias batch effect and patient-bias batch effect. (b) Macenko stain normalization is applied to patches before feature extraction, but the effect is very limited. (c) After applying fmMAP, the batch effect is significantly reduced in both cases.

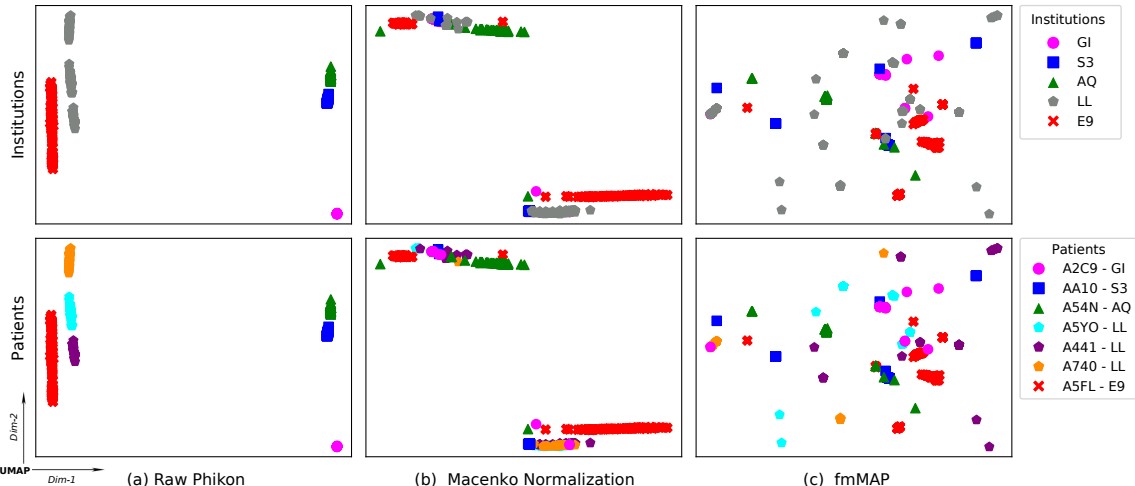

Figure B.6: A UMAP illustration of site-bias effect in a subset of BCSS dataset using FM Phikon (Filiot et al. (2023)). (a) Raw features extracted from FM clearly show the site-bias batch effect and patient-bias batch effect. (b) Macenko stain normalization is applied to patches before feature extraction, but the effect is very limited. (c) After applying fmMAP, the batch effect is significantly reduced in both cases.

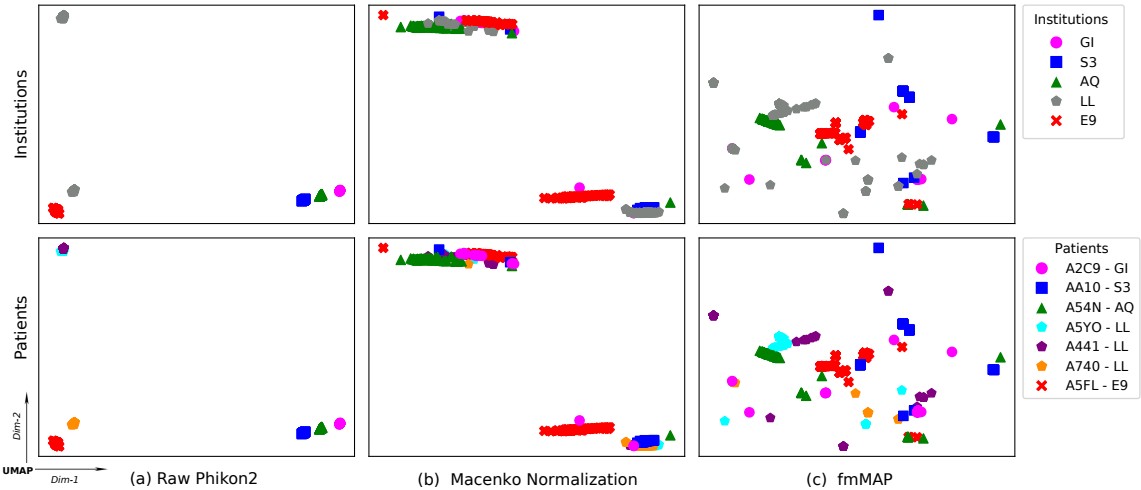

Figure B.7: A UMAP illustration of site-bias effect in a subset of BCSS dataset using FM Phikon-v2 (Filiot et al. (2024)). (a) Raw features extracted from FM clearly show the site-bias batch effect and patient-bias batch effect. (b) Macenko stain normalization is applied to patches before feature extraction, but the effect is very limited. (c) After applying fmMAP, the batch effect is significantly reduced in both cases.

