# OpenReview forum: "fmMAP: A Framework Reducing Site-Bias Batch Effect from Foundation Models in Pathology"
_MICCAI.org/2025/Workshop/COMPAYL — COMPAYL 2025_

### Official Review · Reviewer_jmQk · 2025-07-13
**Simple and Effective Batch Effect Removal for Pathology Foundation Models**

**Rating:** 4
**Confidence:** 4

**Review:**

**Short Summary**

This paper introduces fmMAP, a supervised UMAP-based pipeline to reduce site-specific batch effects in foundation models for pathology. The method significantly improves robustness and tissue classification accuracy across multiple models and datasets, outperforming standard stain normalization. While the approach is simple and experiments are extensive, the lack of publicly available code limits reproducibility and broader impact.

**Strengths**

Addresses a critical and well-documented problem (batch effects) in computational pathology with a novel, simple, and effective solution.

Extensive and thorough experimental validation across multiple foundation models and datasets. Demonstrates clear improvements in robustness.

Strong motivation and clear writing, with comprehensive literature review and citations.

The pipeline is general and can be integrated with various normalization methods.

Results are consistent and robust across different experimental settings.

**Weaknesses**

Reproducibility is limited due to the unavailability of code or a pip package, making it difficult for others to replicate or extend the work. Release the code and ideally provide a pip package to facilitate reproducibility and allow easy evaluation on new foundation models with more normalization methods.

Figures (especially Figures 3 and 4) are hard to interpret and compare across models and normalization methods, limiting accessibility of results. Improvements potentials: Visualizations could be improved by grouping or color-coding by normalization, and maintaining consistent ordering for easier comparison (injection of 1280token variants make it harder). Include plots that average over foundation models or explicitly show the effects of normalization and 1280-token variants, making it easier to assess trends. More explicitly highlight and visualize the drops or gains introduced by stain normalization and fmMAP per model.

Lack of analysis on whether improvements in robustness indices translate to better generalization on unseen, out-of-distribution test sets.

---

### Official Review · Reviewer_fBsv · 2025-07-13
**Trade-off between downstream performance and robustness**

**Rating:** 3
**Confidence:** 4

**Review:**

**Summary**: The manuscript proposes a framework to transform pathology foundation model features to achieve more robust features to batch effects. Using supervised uniform manifold approximation (UMAP) with patch-level task labels to transform the embeddings while keeping the original dimension, the resulting features achieve better performance on the downstream tasks linked to the labels. At the same time, the features are more robust to source-site-prediction.

**Strengths**: The paper explores a simple approach to counteract the observed lack in robustness of pathology foundation model features, an important topic for the community of computational pathology. The tool could be used as preprocessing step before using pre-computed foundation model features for linear probing. The authors plan to release the code.

**Weaknesses**:
* The paper proposes to inform model representations with task knowledge. Consequently, these representations perform better on the downstream tasks when evaluated with linear probe. However, it should also be evaluated whether the representations remain general-purpose. If the representations are only evaluated with respect to one single use case, they should be benchmarked with fine-tuned foundation models, which should also be tested for robustness. My hypothesis would be that improving representations with supervision on one task, necessarily leads to lower performance on other non-related tasks. However, this harms the goal of general-purpose representations.
* The chapter “related work” does not capture the state-of-the-art in stain normalization and stain augmentation well and therefore also only includes outdated methods for stain normalization (Macenko and Reinhard).
* Relying on UMAP representations for claims on robustness is not good practice since UMAP representations can be misleading (Most recent reference among many others: “Stop Misusing t-SNE and UMAP for Visual Analytics”, Jeon, H., et al., arxiv: 2506.08725). In the manuscript, this should be emphasized more, even though I acknowledge that the paper also uses a robustness score.

**Minor issues**:
* Figure 1:
  * add small images of example patches to the picture that the morphological and site-specific differences become more clear
  * change colors such that that same patients and sites have the same colors in the two rows except for the domain LL
  * add a third row including the semantic labels to not only rely on umap clustering
  * stain normalization should as marked as optional
* Figure 3: group the same foundation models visually together to make it easier to grasp the differences between the models. Alternatively, you could separate

---

### Official Review · Reviewer_CsmH · 2025-07-15
**A Framework for Reducing Site Bias in Pathology Foundation Models**

**Rating:** 4
**Confidence:** 5

**Review:**

Summary:
This paper introduces fmMAP, a post-hoc, model-agnostic method for reducing site-related batch effects in foundation model (FM) features from histopathology images. It uses a supervised UMAP transformation to align feature representations based on biological labels (e.g., tissue type) while minimizing confounding site-specific variation. The method is evaluated across multiple foundation models and two public datasets.

Strengths:
The paper addresses an important and underexplored problem: site bias in FM features, which undermines generalizability and a post-hoc correction without retraining is a practical strength.The method is conceptually simple but well-motivated and empirically testing across 8 FMs and different stain normalization settings. Preserves biological information while reducing site-specific clustering, as demonstrated by classification accuracy and robustness index.

Weaknesses:
The evaluation is limited to patch-level tissue and site classification. The method would be more compelling if tested on clinical downstream tasks (e.g., biomarker prediction, prognosis). Figures are hard to interpret due to unclear legends and lack of quantitative cluster metrics. No comparison is made to recent related work on FM bias quantification or contrastive learning-based harmonization.
The approach relies on patch-level labels during UMAP training, which limits scalability to label-scarce scenarios.